# Livestock Depredation by Large Carnivores and Human–Wildlife Conflict in Two Districts of Balochistan Province, Pakistan

**DOI:** 10.3390/ani14071104

**Published:** 2024-04-04

**Authors:** Najeeb Ullah, Irum Basheer, Faiz ur Rehman, Minghai Zhang, Muhammad Tayyab Khan, Sanaullah Khan, Hairong Du

**Affiliations:** 1College of Wildlife and Protected Area, Northeast Forestry University, No 26, Hexing Road, Harbin 150040, China; najeebachakzai3@gmail.com (N.U.); dhr9012@163.com (H.D.); 2Key Laboratory of Saline-Alkali Vegetation Ecology Restoration, Ministry of Education, College of Life Science, Northeast Forestry University, Harbin 150040, China; irumbashir598@gmail.com; 3Department of Zoology, Government Superior Science College Peshawar, Peshawar 25000, Pakistan; faiz02140@gmail.com; 4Department of Land, Environment, Agriculture and Forestry, University of Padova, 35131 Padova, Italy; muhammadtayyab.khan@studenti.unipd.it; 5Department of Zoology, University of Peshawar, Peshawar 25000, Pakistan; sanaullahkhan@uop.edu.pk

**Keywords:** predators, livestock, depredation, gray wolf, caracal, Asiatic jackal, striped hyena

## Abstract

**Simple Summary:**

Livestock herding is an essential and time-honored practice in Balochistan, playing a pivotal role in the region’s economy, culture, and way of life. The livestock sector is of utmost important in the province, catering to nearly 20% of the national stock. Over centuries, large predators and their prey species, including livestock, have coexisted in these mountainous landscapes. Consequently, large carnivores are more likely to interact with humans due to their extensive home ranges. This research aims to explore the impact of livestock depredation by large predators on livelihoods and conservation efforts in two districts of Balochistan, Pakistan. A human–carnivore conflict survey was conducted from July to September 2019, gathering data from 311 residents in the selected study area. Large predators in the region preyed on a total of 876 livestock over a year, comprising 560 goats, 292 sheep, 19 cows, and 5 donkeys. The gray wolf emerged as the primary predator, accounting for 66.3% of livestock depredation, which was followed by the caracal (24.3%), Asiatic jackal (8.9%), and striped hyena (0.6%). The economic loss totaled USD 78,694. Notably, 80% of respondents held negative perceptions toward wolves compared to 24.4% for caracals. Only 20.6% of respondents were aware of the importance of conserving carnivores. Livestock depredation by carnivores has fostered negative perceptions among locals toward these animals. There exists a lack of awareness regarding the significance of conserving carnivore species and their ecological roles. It is crucial to raise awareness among communities about the ecological importance of predators like the gray wolf, caracal, Asiatic jackal, striped hyena, and Balochistan black bear through community meetings and educational seminars. Furthermore, providing basic education to herders on effective livestock guarding practices is recommended to mitigate human–carnivore conflicts and promote coexistence between wildlife and local communities in Balochistan.

**Abstract:**

Livestock herding is a vital practice in Balochistan, contributing to the economy and culture. The livestock sector is significant in Balochistan, providing 20% of the national stock. Large predators and their prey species, including livestock, have coexisted in these mountainous landscapes for centuries. The aim of the present research is to investigate the impacts of livestock depredation by large predators on livelihoods and predator conservation in two districts of Balochistan, Pakistan. A human–carnivore conflict survey was conducted from July to September 2019, collecting data from 311 residents in a selected study area. Large predators in the study area preyed on a total of 876 livestock during a one-year period, including 560 goats, 292 sheep, 19 cows, and 5 donkeys. The gray wolf is the leading predator, responsible for 66.3% of livestock depredation, followed by the caracal (24.3%), Asiatic jackal (8.9%), and striped hyena (0.6%). The total economic loss was USD 78,694. Overall, 80% of respondents had a negative perception of wolves compared to 24.4% for caracals. Only 20.6% of respondents knew about the importance of conserving carnivores. Livestock depredation by carnivores in the study area created a negative perception of these animals among people. There is a lack of awareness about the importance of conserving carnivore species and their role in the ecosystem. This lack of understanding has ultimately led to detrimental effects on predator populations. It is imperative to raise awareness among people about the ecological significance of carnivores through community meetings, seminars in educational institutions, and providing basic education to herders about effective livestock guarding practices.

## 1. Introduction

Balochistan is Pakistan’s largest province by territory, where a significant portion of the population engages in livestock rearing across communal and open-access rangelands. Although the province makes up 44% of the total geographical area, it has only 5% of arable land [1]. Balochistan’s livestock sector, a cornerstone of the regional economy and cultural heritage, harbors approximately 20% of the national livestock population [2]. Arid climatic conditions and limited water resources in Balochistan render crop production unreliable, forcing farmers to depend on livestock rearing for sustenance and income. This dependence is further amplified by the growing human population, leading to an increase in livestock production in the province [2].

Human–wildlife conflict results from the rising human population in proximity to wildlife habitats. As the human population increases and the demand for resources grow, the frequency and intensity of such conflicts increases [3]. These conflicts may result when wildlife damage crops or when they threaten, kill or injure people and domestic animals. Growing livestock populations lead to increased resource competition between wildlife and livestock [4]. With more livestock available, carnivorous wildlife species find an abundant and easily accessible prey source, thereby escalating predation on domestic animals [5,6,7]. Expanding livestock numbers often lead to a need for increased grazing pastures and human settlements to support them [8]. However, this process of growth often leads to the fragmentation of ecosystems, wherein once undisturbed natural environments are transformed into agricultural and grazing lands. This phenomenon not only reduces the amount of space available for animals but also leads to habitat overlap between wildlife and humans [9,10]. The consequences are significant, leading farmers to use lethal predator control methods such as trapping, poisoning, or shooting to save their cattle [11,12]. These threaten not only the intended predators but also non-target species, which causes more problems in ecosystem dynamics and biodiversity.

Large predators and their prey species, including livestock, have coexisted in these mountainous landscapes. Protected areas have fostered growth in predator populations. However, this success brings a new challenge: increased overlap between carnivore habitats and human activities, especially along protected area boundaries where livestock graze [13]. In the study area, predators such as wolves, bears, hyenas, jackals and caracals are present. The largest home size recorded range from 259 to 1716 square kilometers for gray wolves [14,15]. Black bears typically have smaller home ranges with estimates around 117 square kilometers [16]. The smallest home ranges for the mentioned species were recorded as 40 to 72 square kilometers and 11.2 to 26.3 square kilometers for hyenas and jackals, respectively [17,18,19,20]. These diverse home range sizes reflect the adaptive strategies of each species in relation to their ecological niches and behaviors.

Livestock depredation by large mammalian predators inflicts significant economic losses on impoverished pastoral communities across Pakistan, jeopardizing village-level food security [21]. Consequently, predation on livestock remains a primary driver of human–carnivore conflict and a challenge for wildlife conservation efforts [22]. This conflict intensifies when livestock predation increases due to a combination of factors: growing predator populations, potentially due to successful conservation measures, and domestic livestock outnumbering wild prey, particularly in mountainous regions where livestock herding is a crucial source of livelihood [23]. A lack of understanding regarding the complex social, economic, and ecological interactions between pastoral communities and threatened wildlife species further exacerbates the conflict [24]. 

Climate change is intensifying conflicts between humans and wildlife by impacting food resources due to temperature rises and precipitation anomalies. This shift in resources, driven by climate change, is leading to increased encounters between wild animals and human settlements, resulting in economic losses, property damage, and threats to both human safety and wildlife populations [25]. As climate change continues to affect ecosystems globally, it is crucial to address these human–wildlife conflicts through comprehensive approaches that consider ecological, social, and climatic contexts. Mitigating climate change impacts and implementing sustainable solutions are essential for the coexistence of humans and wildlife in the long term [26].

We conducted this study to determine out the social and economic importance of livestock herding and livestock depredation as well as the possible causes and effects of livestock depredation on local livelihoods and the conservation of large predators. Our goal was to identify appropriate conservation measures to keep both pastoral activities and the conservation of large predators in balance.

## 2. Methods

### 2.1. Study Area 

The present study was conducted in the Khuzdar (27.5758° N, 66.8082° E) and Lasbela (25.8700° N, 66.7129° E) regions of Balochistan (Figure 1), Pakistan, which possess the most significant species diversity across the province [27]. The elevation ranges varies considerably, ranging from sea level to 1494 m. The area experiences a warm arid climate with distinct summer and winter seasons. Summers are hot and dry, typically lasting from April to October. June is the hottest month with average high temperatures exceeding 32 °C. Winters are mild, spanning from November to March. January is the coolest month with average low temperatures around 19 °C [28]. Precipitation is scarce, with most occurring in July and August. The average annual rainfall is only 3.4 mm [29].

The study area exhibits a rich terrestrial flora, including *Euphorbia neriifolia* (Indian spurgetree) *Caragana polyacantha* (polyacantha), *Convolvulus spinosus* (Ritchak OR Dolako), *Fagonia arabica* (Dhamasa), *Acacia rupestris* (gum acacia), *Capparis aphylla* (karira), bushy and leafy *Salsola* spp., *Olea europaea* (olive), and *Tamarix aphylla* (tamarix) [30]. A diverse mammalian fauna is also present, comprising *Canis lupus* (gray wolf), *Vulpes vulpes griffithii* (hill fox), *Canis aureus* (Asiatic jackal), *Hyaena hyaena* (striped hyena), *Lepus capensis* (cape hare), *Hystrix indica* (porcupine), *Hemiechinus auritus megalotis* (hedgehog), *Capra aegagrus* (*Sindh ibex*), *Ovis vignei cycloceros* (*Afghan urial*), *Gazella benettii* (chinkara), *Felis silvestris* (desert cat), and *Golunda ellioti* (bush rat) [31].

### 2.2. Methods: Human–Carnivore Conflict Survey

A combination of qualitative methods, specifically key informant interviews, and quantitative methods, using structured interviews with a detailed questionnaire, were employed to collect data from the field, as has been implemented in previous studies [13,14,15,16,17,18,19,20,21,22,23,24,25,26,27,28,29,30,31,32]. Structured interviews were conducted with 311 respondents from 40 villages with documented livestock depredation incidents and a high concentration of herding activity. These villages were selected across two southeastern districts of Balochistan based on these criteria. Structured interviews were conducted with 311 participants (aged > 18 years) from July 2019 to September 2019 (details in Annexure 1). 

In the questionnaire surveys, we included the following. (1) The socioeconomic information (name and surname, age, education, livestock types, and number of livestock owned) established a baseline understanding of participants’ livelihood dependence on livestock. (2) In-depth information on livestock depredation events was collected, including frequency of depredation incidents experienced by participants or their communities over the past year. Species of carnivore responsible for the attacks were identified using color photographs presented to participants. This ensured accuracy in predator identification. Number and types of livestock depredated. This provided a clear picture of the economic impact. (3) Lastly, respondents’ attitudes toward carnivore conservation in Pakistan were measured [33]. Discussions following the questionnaire explored participants’ personal experiences with carnivore conflict, including details of personal encounters with carnivore attacks on livestock and observations of predator sightings in their surrounding areas over the past year as per Mishra et al.’s (2003) documentation [34]. This information complemented the data collected in the questionnaire.

#### Sign Survey

To validate the presence and distribution of large carnivores across the study area, we conducted a sign survey following established protocols [35,36]. This survey employed a grid-based sampling design using ArcGIS v10.8 [37], which was used to overlay a grid system with 15 km × 15 km squares across the entire study area (45,123 km^2^). This facilitated a systematic search within manageable sections. Within each grid cell, five random points were chosen within a 50 m radius using a random number generator. These points served as search locations for carnivore signs. At each random point, we searched for potential carnivore signs, including feeding remains (scat), hairs, tracks, footprints, and claw markings on trees. Signs were classified into two categories based on estimated age: fresh (<1 month old) and old (1–12 months old). The age of claw markings was determined by examining the color and regrowth of bark within the gouges. A Global Positioning System (GPS) was used to record the precise location coordinates of each identified sign.

### 2.3. Data Analysis

We used ArcGIS version 10.8 [37] to map the areas of livestock depredation within the designated region. Statistical analysis was conducted to identify the key factors influencing people’s perceptions of carnivores in the study area. Utilizing a binomial logistic model (Appendix A) [38], we examined the data related to human–wildlife conflict. This analysis allowed us to evaluate impact of social factors including education, occupation, economic losses due to depredation, total livestock count, and instances of livestock depredation by the carnivorous predators. Furthermore, we examined ecological factors such as the season and timing of livestock grazing.
(1)$$lm\left(formula=Total\;Depredation\;\sim \;factor\left(Season\right)+factor\left(guarding\right)+factor\left(Total\;livestock\;owned\right)\right)$$

We assessed the relationship between the total number of depredations and various factors, including season, education, timing, and total livestock owned. The model aims to identify potential associations or influences on the total number of livestock depredations.
(2)$$lm\left(formula=Predator\;Perception\;\sim \;factor\left(Education\right)+factor\left(Economic\;loss\;Total\right)+factor\left(Occupation\right)+Total\;livestock\;owned\right)$$

It shows the factors influencing people’s perception of predators. The factors considered include education, economic loss, occupation, and total livestock owned.

## 3. Results

### 3.1. Demographic Profile and Livestock Ownership Patterns among Study Respondents

The majority of the 311 respondents in the study area had limited education with 29.2% illiterate and 49.6% having only primary to middle school levels. Only 15.8% had completed high school, and 5.4% had pursued higher education. A small percentage of respondents were knowledgeable (20.6%) about the importance of conserving carnivore predators. The majority of respondents were involved in rearing livestock. Goats constituted the majority of the livestock with sheep, cows, and donkeys following in that order (Table 1).

### 3.2. Extent and Economic Implications of Livestock Losses Due to Predation among Study Respondents

During the study period, 146 respondents lost a total of 876 livestock in the past year (2019–2020). The predators targeted various livestock species, with goats accounting for the largest portion (63.9%), followed by sheep (33.3%), cows (2.1%), and donkeys (0.6%). The total economic impact of these losses was 78,694 USD. This breakdown to goat losses amounting to 47,060 USD, sheep losses to 24,847 USD, cow losses to 5774 USD, and donkey losses to 1013 USD (Table 2).

### 3.3. Livestock Depredation by Predator Species in the Study Area

The data collected from respondents showed that wolves are the main predator responsible for livestock depredation (66%) followed by caracal (24%) and Asiatic jackal (9%) in the study area. Hyenas are rarely involved (1%) (Table 3) (Figure 2). Black bears were not reported to have preyed on any livestock. 

### 3.4. Perception of Respondents toward Carnivorous Predators in the Study Area

The survey revealed negative perceptions toward some carnivores, particularly wolves (n = 249), caracals (n = 76), and Asiatic jackals (n = 26). Our binomial logistic regression model showed a significant association between economic losses from predation and negative perception toward carnivores (*p* = 0.001). In terms of basic education, respondents with higher education were less likely to report wolf predation, similar to the cases of caracals and Asiatic jackals, when compared to those with no education or basic education (*p* = 0.001), and the same is the case for caracals as well as Asiatic jackals. Interestingly, binomial logistic regression models showed that higher education levels were associated with less negative perceptions of wolves and caracals while also suggesting a link between education and reported predation events (Table 4). This suggests that education may influence how people perceive and report carnivore interactions. Conversely, economic losses from wolf and caracal attacks led to more negative attitudes.

### 3.5. Seasonal Variation in Predation on Livestock 

To explore the relationship between seasonality and predation, the model revealed a statistically significant correlation between seasonality and livestock depredation by various carnivores. Wolf predation was highest in the winter (52%), which was followed by autumn (28%), spring (12%), and summer (8%) (*p* = 0.001). Caracal predation exhibited a peak during the summer (37%), which was followed by spring (29%), winter (20%), and autumn (14%) (*p* = 0.005). Asiatic jackal predation was most prevalent in autumn (61%) and winter (27%) with lower rates in spring (9%) and summer (3%) (*p* = 0.04). Hyena predation occurred predominantly in autumn (60%) and winter (40%) (*p* = 0.04). The findings underscore a direct correlation between livestock depredation and seasonal variations (*p* > 0.001), indicating fluctuating rates of predation depending on species and their behavioral patterns (Figure 3).

### 3.6. Carnivore Species Diversity and Abundance 

The interviews combined with sign observations revealed the presence of six carnivore species in the study area: Asiatic jackal (most frequently sighted, n = 291), caracal (n = 246), gray wolf (n = 174), Balochistan black bear (n = 27), and jungle cat (least frequently sighted, n = 17). This suggests a decreasing frequency of sightings from jackals to jungle cats (Figure 4)

### 3.7. Assessing the Abundance and Rarity of Carnivore Species 

In our surveys, jackals were perceived as the most prevalent (51% categorized them as common), which was followed by caracals (43%), wolves (49%), and striped hyenas (31%). Black bears were seen as relatively rare (3% common), and none of the participants considered the jungle cat to be common. This suggests a decreasing perception of abundance from jackals to jungle cats (Figure 5)

Our sign survey found many pugmarks from Asiatic jackals, caracals and wolves, indicating an abundance of these species, while analysis of pug marks revealed the Asiatic jackal, caracal, and wolf as the most abundant species, while a few pug marks suggested a limited presence of striped hyena. Additionally, hair samples confirmed the presence of the black bear. These findings (detailed in Figure 6) contribute to a more comprehensive understanding of carnivore distribution within the study area (Figure 6).

## 4. Discussion

This study investigated human–wildlife conflict arising from livestock depredation by carnivores in a region heavily reliant on natural resources. As expected, our findings align with previous research indicating that communities with a high dependence on livestock production tend to develop negative perceptions of predators due to predation events [39]. The residents of the present study area predominantly belong to the middle or lower socioeconomic classes, deriving their livelihoods primarily from farming and livestock-related activities. Notably, the study area did not generate any revenue from tourism, and there are limited government regulations in place. The inhabitants of this region are primarily engaged in substantial livestock rearing for both economic and dairy purposes. This practice contributes to the rising livestock trends among the local residents of the study area, which correlates with a study conducted in the Misgar Valley of Hunza [38]. Our findings corroborated previous studies on predator selectivity [32,40,41]. Caracals and jackals primarily preyed on sheep, which was likely due to their smaller size and flocking behavior. Conversely, wolves seemed to target goats more frequently. These observations are consistent with the established notion that predators exhibit size-selective predation [24,25,26,27,28,29,30,31,32,33,34,35,36,37,38,39,40,41,42]. Notably, most depredation events involved livestock grazing unguarded, highlighting the importance of active herding practices for mitigating predation [43].

In our study, the residents of the study area experienced the greatest economic losses due to wolves, which was followed by caracals and Asiatic jackals. While there were only a few reported cases of depredation by hyenas, the cumulative economic loss caused by all predators amounted to USD 78,694. Our study revealed a negative correlation between educational attainment and the perception of wolves and caracals as threats to livestock. This suggests that respondents with lower education levels were more likely to hold negative views toward these carnivores. This negativity likely stems from the high incidence of livestock depredation. The negative perception toward carnivores, as confirmed in the present study and previous literature [1,13,44], poses a significant threat to predator populations.

In the present study, only 20.6% of respondents were knowledgeable about the importance of conserving predators such as the wolf, caracal, Asiatic jackal, and hyena. The majority of respondents, however, seemed to be unaware of the importance of conservation. Predators such as the gray wolf, Asiatic jackal, caracal, striped hyena, black bear, and jungle cat have been recorded in Balochistan previously [45]. In the present study, the presence of these carnivore species was confirmed during the study period through direct sightings by respondents and researchers as well as by observing signs such as pug marks, feces, and hair. These data suggest that the gray wolf and Asiatic jackal are the most common carnivores in the study area and are responsible for the majority of livestock depredation. 

Further research is needed to assess the current status of all carnivore species in the study area and to develop effective management strategies to mitigate human–wildlife conflict. Interviews revealed that most respondents lack well-developed protective strategies against carnivores. This includes a shortage of trained guardian dogs, inadequate confinement of livestock at night, and unsupervised grazing in large open fields. These practices likely contribute to livestock predation. Guardian dogs can play a crucial role in protecting livestock from predation. These dogs are used worldwide to reduce livestock depredation, ultimately decreasing the need for lethal predator control and benefiting conservation efforts [46]. Studies suggest that the presence of guarding dogs around the herd can prevent attacks by carnivores and reduce surplus killing [47]. 

The economic burden of predator conflicts on villagers is recognized, and compensation is identified as a beneficial mechanism for alleviating this burden [48]. A study in Botswana’s Okavango Delta examined livestock depredation events and compared them with the country’s compensation program. The program’s limitations in sustainability and claim verification were acknowledged. However, the researchers recommend enhancing the program, not abandoning it. Improvements should focus on ensuring timely reporting and thorough investigations of depredation events to improve effectiveness [43]. To address this issue, compensation for livestock depredation is proposed as a means to mitigate costs. Such programs are designed not only to provide financial support to producers but also to reduce animosity toward predators, contributing to the overall conservation effort [49]. We recommend parallel efforts to promote coexistence with predators. Targeted training and awareness programs within the study area can educate residents on livestock protection strategies like proper enclosures and fencing. Community meetings, educational seminars, and campaigns can raise awareness about the importance of coexistence. Exploring alternative income opportunities for herders, such as sustainable agriculture, beekeeping, and environmentally friendly practices, can further reduce reliance on livestock and potential human–wildlife conflict. Finally, investigating the feasibility of compensation programs similar to those in other regions can provide financial support for depredation losses and discourage retaliatory killings.

## 5. Conclusions

The predation of livestock by carnivorous predators in the Lasbela and Khuzdar regions of Balochistan has significant economic implications for local communities. To mitigate this issue, it is imperative to increase awareness among the local population about the ecological significance of predators like the gray wolf, caracal, Asiatic jackal, striped hyena, and Balochistan black bear. This awareness can be raised through community meetings, seminars in educational institutions, and targeted educational campaigns emphasizing the importance of coexisting with these species. Moreover, educating herders on effective livestock safeguarding measures is crucial to ensure the protection of their animals. Exploring alternative livelihood options for herders, such as promoting sustainable agriculture, beekeeping, and other environmentally friendly practices, can provide them with alternative sources of income. Additionally, it could also be worthwhile to explore compensation programs for depredation events similar to those implemented in other regions that also discourage people from killing carnivores.

## Figures and Tables

**Figure 1 animals-14-01104-f001:**
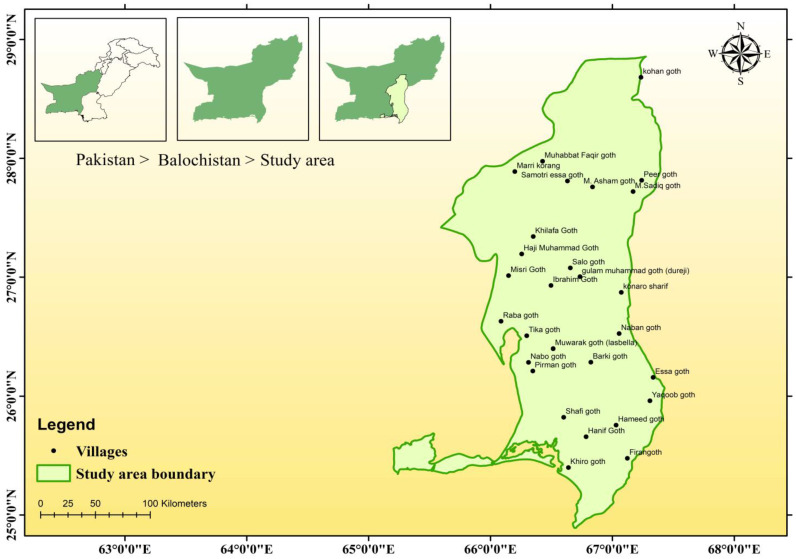
Study area map showing survey locations in southeast Balochistan province, Pakistan.

**Figure 2 animals-14-01104-f002:**
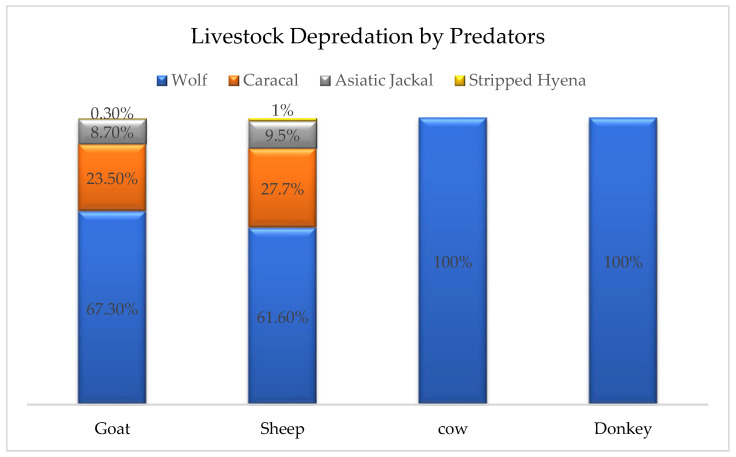
Livestock depredation by wolf, caracal, Asiatic jackal, and striped hyena.

**Figure 3 animals-14-01104-f003:**
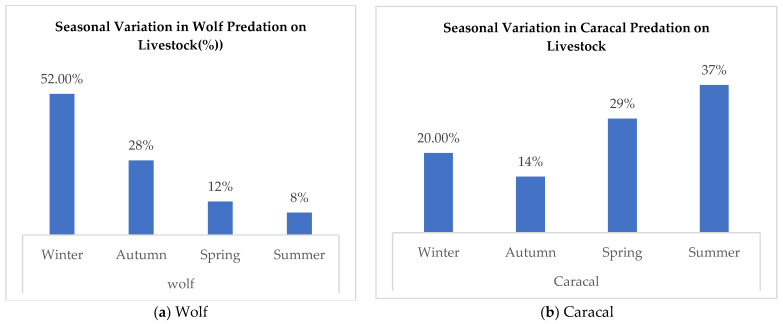
Seasonal variation in predation on livestock.

**Figure 4 animals-14-01104-f004:**
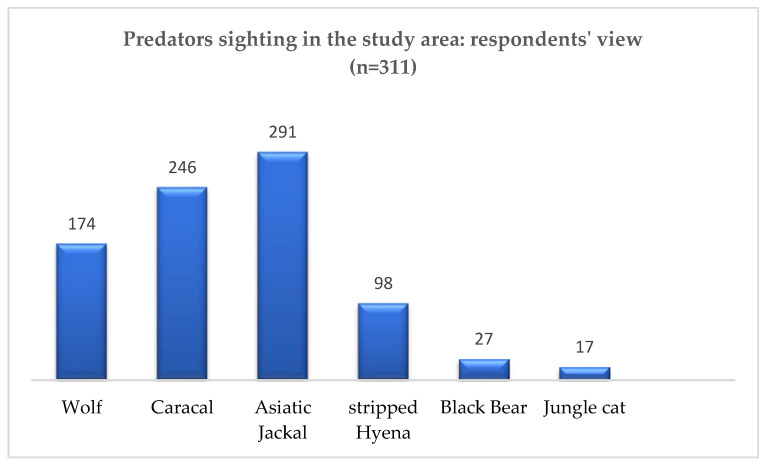
Carnivore species diversity and abundance.

**Figure 5 animals-14-01104-f005:**
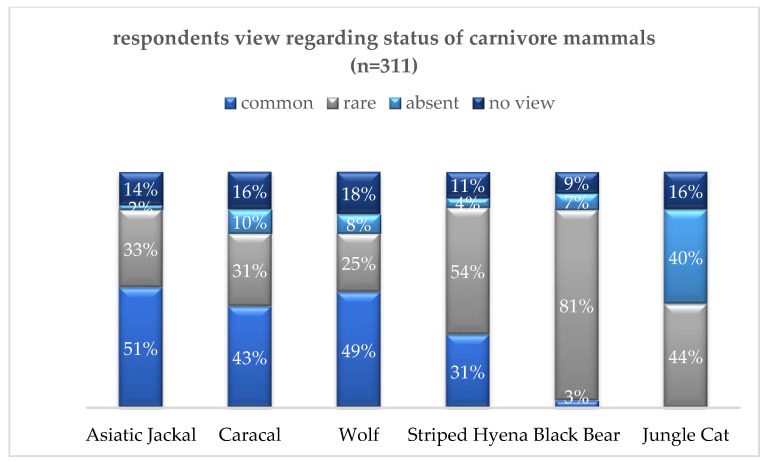
Abundance and rarity of predators.

**Figure 6 animals-14-01104-f006:**
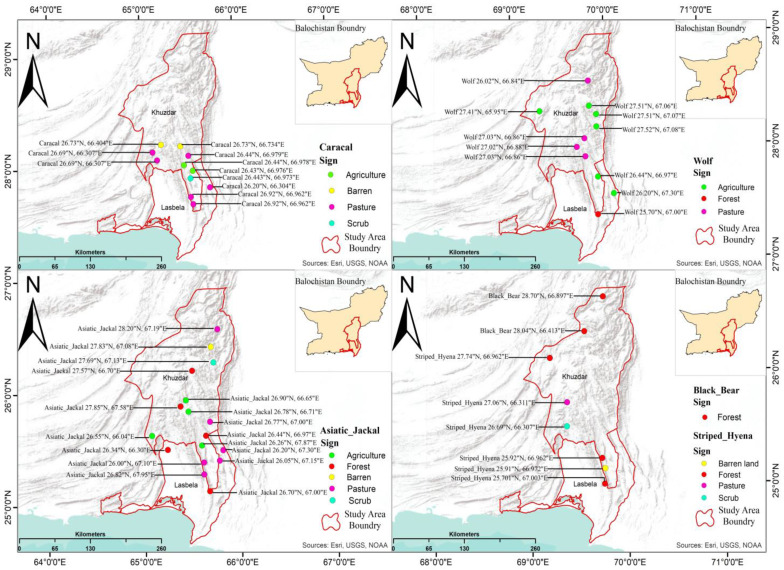
Map showing distribution of studied carnivores.

**Table 1 animals-14-01104-t001:** Demographic profile and livestock ownership patterns among the respondents in study area.

Characteristics	Sub Category	Numbers	Percentage
Questionnaire		311	100
Education	Illiterate	91	29.2
	Basic education (primary–middle)	154	49.6
	High school (matric–intermediate)	49	15.8
	Higher education	17	5.4
Occupation	Shepherd	174	56
	Farmers	116	37.2
	Employees	16	5.1
	student	5	1.6
Knowledge about carnivore conservation	Yes	67	20.6
	No	247	79.4
Livestock holders		252	81
		5720	100
Total numbers of livestock	Goats	3385	59.2
	Sheep	2010	35.1
	Cows	250	4.4
	Donkeys	75	1.3
Livestock owned	0	58	18.7
	≤10	55	21.7
	11–20	67	26.4
	21–30	56	22.1
	31–40	54	21.3
	≥40	21	8

**Table 2 animals-14-01104-t002:** Extent and economic implications of livestock losses due to predation among study respondents.

Species of Livestock	Numbers of Species Preyed by Predator	Unit Price of Livestock (PKR)	Total Price in PKR	Total Price in USD
Goat	560	25,000	14,000,000	47,060
Sheep	292	25,000	7,300,000	24,847
Cow	19	90,000	1,710,000	5774
Donkey	5	80,000	300,000	1013
Total	876		23,310,000	78,694

**Table 3 animals-14-01104-t003:** Livestock depredation by the carnivores in the study area.

Predator	Livestock Species				Total
	Goat	Sheep	Cow	Donkey	
Wolf (*Canis lupus*)	377	180	19	5	581
Caracal (*Caracal Caracal*)	132	81	0	0	213
Asiatic jackal (*Canis aureus*)	49	28	0	0	77
Striped hyena (*Hyaena hyaena*)	2	3	0	0	5
Balochistan black bear (*Ursus thibetanus gedrosianus*)	0	0	0	0	0

**Table 4 animals-14-01104-t004:** Perception of respondents toward predators in the study area.

Predator	Respondents’ Perception (n)	
	More Dangerous	Less Dangerous
Wolf	249	62
Caracal	76	235
Asiatic jackal	26	285
Hyena	0	311
Black bear	0	311
Jungle cat	0	311

## Data Availability

Data are contained within the article.

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
