# Peer review of "Livestock Depredation by Large Carnivores and Human–Wildlife Conflict in Two Districts of Balochistan Province, Pakistan"

_animals, 2024, doi:10.3390/ani14071104_

Round 1

Reviewer 1 Report (Previous Reviewer 1)

Comments and Suggestions for Authors

The new version submitted is well written and good improvements had been made to the original version. Anyway some changes have to be performed in order to go ongoing with the publication.

Abstract.

It's still too longer. Please reformule from line 31 to 35 searching to  synthesize the concepts that, in any case, are already present and discussed in the following sections.

Line 20: here there is the aim of the work, please search to underline it for example starting the sentence: "for these reasons, the aim of the present research is....".

Introduction

Line 71. After concluding these points, I suggest the Authors also to include in this section the important link between wildlife/livestock competition and climate change. I know that it's not the topic of the work, but I think that the reader must be informed also about this problem. In fact, knowing the exitance of the conflit between livestock and wildlife in certain areas of the world, the rising of the temperatures or the anomaly of the precipitations can negatively influence the relationship caused by  the lack of food resources and the competition for them between not only animals but also humans. 

I advice the Authors to speak also about the use of news technologies for the correct comprehension of animal behavior and diseases, with an important influence on Public Health. Please see the suggestion given in the section Material. 

Material

Line 171. You use GPS signal for the position. It will be interesting if you improve the introduction speaking about the importance of these type of data for monitoring wildlife and receive important information about the ecology of species. In particular, the recent use of GIS technology and Remote Sensing, camera traps and other devices could represent a valid instrument for the study of animal behaviour and pathology. Please see the following articles: https://doi.org/10.3390/rs12213542;  

https://doi.org/10.1111/csp2.239). 

Conclusions

This part is still the same of the old version. Please, give pratic solutions adoptable in the study area and describe for exemple the situation in other area of the Word in wich solution adopted had help the resolution of the confilt. Not be too generic here. 

Author Response

Response to reviewer 1 comment

Note: Response to reviewer 1 is highlighted in Yellow in revised file.

The new version submitted is well written and good improvements had been made to the original version. Anyway some changes have to be performed in order to go ongoing with the publication.

Abstract.

It's still too longer. Please reformule from line 31 to 35 searching to  synthesize the concepts that, in any case, are already present and discussed in the following sections.

Thank you, abstract has been revised

Line 20: here there is the aim of the work, please search to underline it for example starting the sentence: "for these reasons, the aim of the present research is....".

Thank you, the correction has been made.

Introduction

Line 71. After concluding these points, I suggest the Authors also to include in this section the important link between wildlife/livestock competition and climate change. I know that it's not the topic of the work, but I think that the reader must be informed also about this problem. In fact, knowing the exitance of the conflit between livestock and wildlife in certain areas of the world, the rising of the temperatures or the anomaly of the precipitations can negatively influence the relationship caused by  the lack of food resources and the competition for them between not only animals but also humans. 

I advice the Authors to speak also about the use of news technologies for the correct comprehension of animal behavior and diseases, with an important influence on Public Health. Please see the suggestion given in the section Material.

Thank you, the introduction section has been revised and all the required information has been incorporated

Material

Line 171. You use GPS signal for the position. It will be interesting if you improve the introduction speaking about the importance of these type of data for monitoring wildlife and receive important information about the ecology of species. In particular, the recent use of GIS technology and Remote Sensing, camera traps and other devices could represent a valid instrument for the study of animal behaviour and pathology. Please see the following articles: https://doi.org/10.3390/rs12213542;  

https://doi.org/10.1111/csp2.239). 

Thank you, this section has been revised.

Conclusions

This part is still the same of the old version. Please, give pratic solutions adoptable in the study area and describe for exemple the situation in other area of the Word in wich solution adopted had help the resolution of the confilt. Not be too generic here. 

Thank you, the conclusion has been revised

Reviewer 2 Report (Previous Reviewer 2)

Comments and Suggestions for Authors

It was nice to see a new version of this paper, and all the improvements that the authors have made to it compared to the first version. It is a lot more clear now, however, further major improvements are still needed before it can be considered for publication. The English language needs improvements, and I strongly recommend the authors find a native English speaker to proof read the document, preferably somebody who has experience with scientific writing. I think there are commercial services for this available as well. I also have some concerns with the description of the methods, the models used, and the discussion in general. Please see my comments below.

Abstract

Retaliatory killing was not investigated in this study, and cannot be stated as a fact here.

Introduction.

Page 2, lines45-47. Livestock herding …… national stock. This is fine, but there is a bit of repetition between the sentences. The first one already mentions that livestock herding is important, which is then repeated in the second sentence.

Page 2, line 48. An area cannot have physiological conditions.

Page 2, line 78. human livestock is not a correct term. Livestock cannot be human.

Page 2, line 79. This is not the right place to start being specific about the study area. An introduction should funnel the reader from a broad topic into the specifics of the study that is being done. At this stage of the introduction, the authors are still talking broadly about human-wildlife conflict, which is fine at this point of the intro. But adding this specific information about the specific study site here is not the right location. This info should either be broadened to talk about predators in Pakistan in general, or this info should be moved to the study site description.

Page 2, line 92. What are conservation rules? Needs rewording or clarification

Page 3, line 99. we could come up with, to devise would be a better term here.

Methods.

Figure 1. It is still not clear to me where in Balochistan the study area is, this needs clarification in the little map in the top left corner.

Page 3, line 108 and 110 and 112. What average is used here to indicate the temperature and rainfall? Needs clarification.

Page 3, lines 113 -115. Please use Latin names everywhere, do not use a mix of common and Latin names. Or use common names followed by Latin names, that is fine as well, but be consistent with which form is used. Given for the fauna the authors use common name (Latin name), I would recommend using that for flora as well.

Page 4, line 128 similar to what other researchers have done earlier is not good language in a paper. Please re-phrase.

Page 4, line 132. we . adult person. That is not true, as the authors just stated that they interviews 311 individuals, please clarify.

Page 4, human-carnivore conflict survey. This section is good, but what I am missing is: how were participants identified for inclusion in the survey? Why those 40 villages, and not others (unless there are only 40 villages in the study area, but then that needs to be stated)? 2) livestock depredation by carnivores ok, but what information was recorded? A simple yes/no, or numbers that were taken by predators? Also, in the results, things are reported on that were not mentioned as being part of the survey, leaving me confused where the info is coming from. If the information is part of the results/analysis, it needs to be mentioned as being part of the survey. All needs clarification. Also, if discussions centered on carnivore sightings and encounters, that also needs to be added to the list of things included in the survey.

Page 4, line 144. areas . questionnaire survey. Is that the survey just described, or a separate one? Needs clarification.

Page 4, line 146-147. we performed . Local shepherd. This needs clarification/detail. It does not make sense to state that one of the methods used to do the sign survey is by doing a sign survey.

Page 4, line 194. Not sure what the authors mean when they say they select 5 random points within a 50m radius. A 50m radius of what? Needs clarification.

Page 4, line 156-157. we used . region. This is not a statistical analysis, and the use of software should really be mentioned where it is talked about. This sentence should move to the relevant spot under sign survey.

Page 4, line 159. Multinomial regression cannot be linear.

Page 4/5. Data analysis section. This section needs work. The authors should be really clear about what they did, and how they did it. Where did this data come from? That needs to explicitly stated. A very clear description is needed of which models were used, and why. A vague description like this analysis allowed .. predators is not enough. Did the authors add every single individual explanatory variable in the model? Or combinations? That is not clear, and needs to be stated. Then, two models are suddenly mentioned, but there is no description leading up to them, and only vague, unclear sentences following them. There is also no description of what the authors aim to do with these models, other than assess the relationship between -dependent variable- and -explanatory variables-. How do the authors aim to do this? That needs to be added. Please clarify this whole section. It also needs to state which software was used for the analysis.

Results

Throughout this whole section (also check rest of the document) carnivorous predation is not a correct term. Better to just use predation.

Table 1. The percentages and numbers are not clear to me. The overall total of respondents is 311, at 100%, I get that. But then total numbers in categories are less than 311, which I can understand if not everybody answers every question. But then how can percentage totals in categories be higher than 100? For example, if 67 (21.5%) have knowledge about carnivore conservation, and 98 (81.6%) do not, then how come that for the total of that category 165 = 103%? Same in other categories. Please double check all numbers and make them add up. Also, be very clear on what percentage is reported, and add that to the description of the the Table. Percentage of the whole, or the percentage of each category?

Table 2. Please double check spelling and grammar. Carnivores are unlikely to pray, etc.

Page 6, header livestock depredation by carnivores in the study area This is confusing, as that was just covered in the previous section. Please re-name.

Figure 2. Should not go to 120% Why do sheep only go up to 60%? Needs explanation or adjustment.
All figures: the y-axis should not have decimal points if they are not used.

Page 7, section 3.4. This whole section is very unclear. I assume the authors are trying to report the results from the models they ran, but they do not make this clear here. The authors need to be very systematic when reporting results. They need to state what they are reporting on, and what they found. Also, using significance in multinomial models is very unreliable, and really does not say all that much about the data.

Page 8, section 3.5. Wolves cannot prey in winter twice.

Section 3.4 and 3.5. As I suggested in the last review, it is much better, and I strongly recommend re-doing this analysis. It would be better to come up with a hypothesis of which variables potentially influence the dependent variable, and then run different models, testing the different hypothesis. Model ranking (for example AIC) can then be used to pick the model that best describes the data, and the effect of the explanatory variables can then be presented in the results.

Page 9, line 241. Feces and hair are not mentioned as being part of the sign survey in the methods. If they were, then that needs to be added to the methods.

Figure 4 and figure 5 and figure 6. Please re-phrase the description to be more to-the-point and informative. Figure 5 should not go to 120%.

Page 10, line 258-263. Why is all of this repeated here again? This was already reported on earlier, and is not necessary, please delete, or integrate with previous sections.

Discussion

The discussion has improved compared to the previous version, however, it still needs a lot of work to improve further before it can be published. The discussion needs a lot more structure, based on own results compared and contrasted with existing literature. It also needs much clearer indication whether a point is made based on the authors own results or on existing literature (which then needs to be referenced).

In the last bit of the discussion, only compensation is put forward as a solution. However, there are many different other solutions that could be used that would protect livestock, such as livestock guardian dogs, livestock management etc. They are all very effective and viable options for the region, why are they not discussed/included?

Page 11, line 279-280. the spatial .. predator type. The authors cannot state this about their study area, as they did not themselves investigate half of the factors mentioned here.

Page 11, line 289-291. prey preference jackals It is unclear how optimal foraging would explain prey preference in this case.

Page 11, line 291-292. The remark about guardian dogs is very interesting, and warrants further discussion, but it is very out-of-place here.

Page 12, line 315-116. Figure 6 does not show anything about retaliatory killing, nor does this study provide any information on that. Please re-phrase.

Table 5 belongs in the results, not the discussion

Conclusions

Page 13, line 364-365. Retaliatory killing was not part of this study, and cannot be included in a statement like that.

The conclusions cannot add new recommendations like they are now. These recommendations should be part of the discussion, not the conclusion. The conclusion is intended to help the reader understand why your research should matter to them after they have finished reading the paper. A conclusion is not merely a summary of your points or a re-statement of your research problem but a synthesis of key points. It cannot add new points though.

Comments on the Quality of English Language

English has improved compared to the previous version, but still needs a lot of further improvements. Grammar is used incorrectly and emotive language is often used. 

Author Response

Response to reviewer 2 comment

Note: Response to reviewer 2 is highlighted in green in revised file.

Comments and Suggestions for Authors

It was nice to see a new version of this paper, and all the improvements that the authors have made to it compared to the first version. It is a lot clearer now, however, further major improvements are still needed before it can be considered for publication. The English language needs improvements, and I strongly recommend the authors find a native English speaker to proof read the document, preferably somebody who has experience with scientific writing. I think there are commercial services for this available as well. I also have some concerns with the description of the methods, the models used, and the discussion in general. Please see my comments below.

Response: Thank you for your thoroughly review. The manuscript has been revised and the issues are resolved.

Abstract

Retaliatory killing was not investigated in this study, and cannot be stated as a fact here.
Thank you, the abstract has been rewritten, and unnecessary details have been omitted.

Introduction.

Page 2, lines45-47. ‘Livestock herding …… national stock’. This is fine, but there is a bit of repetition between the sentences. The first one already mentions that livestock herding is important, which is then repeated in the second sentence.

Thank you for your thorough review. The manuscript has been revised and carefully checked to ensure there is no repetition of the same information.

Page 2, line 48. An area cannot have physiological conditions.

The correction has been made.

Page 2, line 78. ‘human livestock’ is not a correct term. Livestock cannot be human.

Thank you, the correction has been made.

Page 2, line 79. This is not the right place to start being specific about the study area. An introduction should funnel the reader from a broad topic into the specifics of the study that is being done. At this stage of the introduction, the authors are still talking broadly about human-wildlife conflict, which is fine at this point of the intro. But adding this specific information about the specific study site here is not the right location. This info should either be broadened to talk about predators in Pakistan in general, or this info should be moved to the study site description.

The introduction has been structured to ensure a smooth transition from a general discussion of human-wildlife conflict to the specifics of the study being conducted.

Page 2, line 92. What are conservation rules? Needs rewording or clarification

Thank you for your feedback. The sentences have been revised to enhance clarity.

Page 3, line 99. ‘we could come up with’, to devise would be a better term here.

Thank you for your feedback. The sentences have been revised to enhance clarity. “We did this to devise the proper conservation measures to balance pastoral activities and conservation of large predators”

Methods.

Figure 1. It is still not clear to me where in Balochistan the study area is, this needs clarification in the little map in the top left corner.

Page 3, line 108 and 110 and 112. What average is used here to indicate the temperature and rainfall? Needs clarification.

Thank you, the sentence has been modified to enhance clarity.

Page 3, lines 113 -115. Please use Latin names everywhere, do not use a mix of common and Latin names. Or use common names followed by Latin names, that is fine as well, but be consistent with which form is used. Given for the fauna the authors use common name (Latin name), I would recommend using that for flora as well.

Thank you for your feedback. The entire manuscript has been revised and checked for scientific names.

Page 4, line 128 ‘similar to what other researchers have done earlier’ is not good language in a paper. Please re-phrase.

Thank you, the sentence has been rephrased

Page 4, line 132. ‘we …. adult person’. That is not true, as the authors just stated that they interviews 311 individuals, please clarify.

Thank you for your feedback. The methodology section has been updated accordingly, addressing the noted correction.

Page 4, ‘human-carnivore conflict survey’. This section is good, but what I am missing is: how were participants identified for inclusion in the survey? Why those 40 villages, and not others (unless there are only 40 villages in the study area, but then that needs to be stated)? 2) livestock depredation by carnivores – ok, but what information was recorded? A simple yes/no, or numbers that were taken by predators? Also, in the results, things are reported on that were not mentioned as being part of the survey, leaving me confused where the info is coming from. If the information is part of the results/analysis, it needs to be mentioned as being part of the survey. All needs clarification. Also, if discussions centered on carnivore sightings and encounters, that also needs to be added to the list of things included in the survey.

Thank you, the methodological section has been revised, enhancing clarity and detail in the updated manuscript.

Page 4, line 144. ‘areas …. questionnaire survey’. Is that the survey just described, or a separate one? Needs clarification.

Thank you, the methodological section has been revised, enhancing clarity and detail in the updated manuscript.

Page 4, line 146-147. ‘we performed …. Local shepherd’. This needs clarification/detail. It does not make sense to state that one of the methods used to do the sign survey is by doing a sign survey.

Thank you, the methodological section has been revised, enhancing clarity and detail in the updated manuscript.

Page 4, line 194. Not sure what the authors mean when they say they select 5 random points within a 50m radius. A 50m radius of what? Needs clarification.

Thank you, the methodological section has been revised, enhancing clarity and detail in the updated manuscript.

Page 4, line 156-157. ‘we used …. region’. This is not a statistical analysis, and the use of software should really be mentioned where it is talked about. This sentence should move to the relevant spot under ‘sign survey’.

Thank you, the methodological section has been revised, enhancing clarity and detail in the updated manuscript.

Page 4, line 159. Multinomial regression cannot be linear.

  1. If your dependent variable has 2 factors then binomial logistics will be applied. so, we have two factors negative perception and positive perception of the people. another reason is we can not access the perception with a single variable. So, we take all the possible variables that reflect the perception of the people about wildlife

Page 4/5. Data analysis section. This section needs work. The authors should be really clear about what they did, and how they did it. Where did this data come from? That needs to explicitly stated. A very clear description is needed of which models were used, and why. A vague description like ‘this analysis allowed ….. predators’ is not enough. Did the authors add every single individual explanatory variable in the model? Or combinations? That is not clear, and needs to be stated. Then, two models are suddenly mentioned, but there is no description leading up to them, and only vague, unclear sentences following them. There is also no description of what the authors aim to do with these models, other than ‘assess the relationship between -dependent variable- and -explanatory variables-. How do the authors aim to do this? That needs to be added. Please clarify this whole section. It also needs to state which software was used for the analysis.

Thank you for your comprehensive review. We have revised the analysis section accordingly, ensuring clarity and removing any ambiguity from the manuscript

We have used a binomial logistic model. In this model, we access the different variables that respond to the dependent (perception) variable. The idea behind the model is to access the perception of the people about wildlife (+,-) with different response variables (education of the people, property owned by the people, etc) in this way the model is evaluated.

Why do we apply the model?

  1. If your dependent variable has 2 factors then binomial logistics will be applied. so, we have two factors negative perception and positive perception of the people. another reason is we can not access the perception with a single variable. So, we take all the possible variables that reflect the perception of the people about wildlife.

Results

Throughout this whole section (also check rest of the document) ‘carnivorous predation’ is not a correct term. Better to just use ‘predation’.

Thank you. The manuscript has been carefully reviewed for mentioned words, and all incorrect terms have been removed in the revised version

Table 1. The percentages and numbers are not clear to me. The overall total of respondents is 311, at 100%, I get that. But then total numbers in categories are less than 311, which I can understand if not everybody answers every question. But then how can percentage totals in categories be higher than 100? For example, if 67 (21.5%) have knowledge about carnivore conservation, and 98 (81.6%) do not, then how come that for the total of that category 165 = 103%? Same in other categories. Please double check all numbers and make them add up. Also, be very clear on what percentage is reported, and add that to the description of the the Table. Percentage of the whole, or the percentage of each category?

Thank you, the figures and tables are updated.

Table 2. Please double check spelling and grammar. Carnivores are unlikely to pray, etc.

Thank you, the correction has been made.

Page 6, header ‘livestock depredation by carnivores in the study area’ This is confusing, as that was just covered in the previous section. Please re-name.

Thank you, the correction has been made.

Figure 2. Should not go to 120% Why do sheep only go up to 60%? Needs explanation or adjustment.
All figures: the y-axis should not have decimal points if they are not used.

Thank you, the figures and tables are updated.

Page 7, section 3.4. This whole section is very unclear. I assume the authors are trying to report the results from the models they ran, but they do not make this clear here. The authors need to be very systematic when reporting results. They need to state what they are reporting on, and what they found. Also, using significance in multinomial models is very unreliable, and really does not say all that much about the data.

Thank you, the whole section has been revised.

Page 8, section 3.5. Wolves cannot prey in winter twice.

Thank you, the correction has been made.

Section 3.4 and 3.5. As I suggested in the last review, it is much better, and I strongly recommend re-doing this analysis. It would be better to come up with a hypothesis of which variables potentially influence the dependent variable, and then run different models, testing the different hypothesis. Model ranking (for example AIC) can then be used to pick the model that best describes the data, and the effect of the explanatory variables can then be presented in the results.

The analysis section has been revised and all the ambiguity has been removed

Page 9, line 241. Feces and hair are not mentioned as being part of the sign survey in the methods. If they were, then that needs to be added to the methods.

Thank you, the correction has been made.

Figure 4 and figure 5 and figure 6. Please re-phrase the description to be more to-the-point and informative. Figure 5 should not go to 120%.

Thank you, all the figures are updated.

Page 10, line 258-263. Why is all of this repeated here again? This was already reported on earlier, and is not necessary, please delete, or integrate with previous sections.

Thank you, the correction has been made.

Discussion

The discussion has improved compared to the previous version, however, it still needs a lot of work to improve further before it can be published. The discussion needs a lot more structure, based on own results compared and contrasted with existing literature. It also needs much clearer indication whether a point is made based on the authors’ own results or on existing literature (which then needs to be referenced).

In the last bit of the discussion, only compensation is put forward as a solution. However, there are many different other solutions that could be used that would protect livestock, such as livestock guardian dogs, livestock management etc. They are all very effective and viable options for the region, why are they not discussed/included?

The discussion section has been revised

Page 11, line 279-280. ‘the spatial ….. predator type’. The authors cannot state this about their study area, as they did not themselves investigate half of the factors mentioned here.

The discussion section has been revised

Page 11, line 289-291. ‘prey preference … jackals’ It is unclear how optimal foraging would explain prey preference in this case.

The discussion section has been revised

Page 11, line 291-292. The remark about guardian dogs is very interesting, and warrants further discussion, but it is very out-of-place here.

The discussion section has been revised

Page 12, line 315-116. Figure 6 does not show anything about retaliatory killing, nor does this study provide any information on that. Please re-phrase.

The discussion section has been revised

Table 5 belongs in the results, not the discussion

Table shows the comparison of the presence of studied carnivores: previous literature, reported by respondents, and our own field observations

This table contains information derived from comparisons with literature, which is why it was included in the discussion section, while Figures 6 display the coordinates/location from which the pug marks were collected

Conclusions

Page 13, line 364-365. Retaliatory killing was not part of this study, and cannot be included in a statement like that.

The conclusions cannot add new recommendations like they are now. These recommendations should be part of the discussion, not the conclusion. The conclusion is intended to help the reader understand why your research should matter to them after they have finished reading the paper. A conclusion is not merely a summary of your points or a re-statement of your research problem but a synthesis of key points. It cannot add new points though.

Thank you, the conclusion has been revised.

Round 2

Reviewer 2 Report (Previous Reviewer 2)

Comments and Suggestions for Authors

The paper has improved compared to the previous version that I have reviewed a while ago. However, I am a little disappointed that some key issues that I have raised a few times had not been addressed at all, and remain an issue. I have made comments directly on the pdf that was submitted by the authors this time, which I have attached to my review, so please see the pdf and its comments for my full review - I have also indicated which comments had not been addressed. Please note in some cases I have crossed out (small) sections of text and in the comment just added the correction of what I think the text should read - this is mainly in cases of English language corrections. 

Comments on the Quality of English Language

The English language has improved, but still needs further improvements. I have made suggestions for improvement in the document, but it could still really benefit from a proper proof-read by a professional language editing service. 

Author Response

I have revised my manuscript based on the reviewers' comments. I have highlighted all the changes and also added comments addressing the questions/objections raised by the reviewers within the manuscript.

This manuscript is a resubmission of an earlier submission. The following is a list of the peer review reports and author responses from that submission.

Round 1

Reviewer 1 Report

Comments and Suggestions for Authors

The manuscript is well written, anyway some changes have to be performed in order to go ongoing with the publication. 

Abstract.

It's too longer. Please reformule from line 31 to 35 searching to  synthesize the concepts that, in any case, are already present and discussed in the following sections.

Line 20: here there is the aim of the work, please search to underline it for example starting the sentence: "for these reasons, the aim of the present research is....".

Introduction

Line 58. Please insert references. Moreover, you say that increasing livestock populations leads to increased resource competition between wildlife and livestock. I think this is a general affirmation, and it's true. But, if possible, delineate this type of situation in the context in which your research is working on, also adding references to possible previously studies. 

Line 71. After concluding these points, I suggest the Authors also to include in this section the importnat link between wildlife/livestock competition and climate change. I know that it's not the topic of the work, but I think that the reader must be informed also about this problem. In fact, knowing the exitance of the conflit between livestock and wildlife in certain areas of the world, the rising of the temperatures or the anomaly of the precipitations can negatively influence the relationship caused by  the lack of food resources and the competition for them between not only animals but also humans. 

I advice the Authors to speack also about the use of news technologies for the correct comprehension of animal behavior and diseases, with an important influence on Public Health. Please see the suggestion given in the section Material. 

Material

I advice you to insert the figure 1 in the chapter "2.1. Study Area". 

Line 171. You use GPS signal for the position. It will be interesting if you improve the introduction speacking about the importance of these type of data for monitoring wildlife and receive important information about the ecology of species. In particular, the recent use of GIS technology and Remote Sensing, camera traps and other devices could represent a valid instrument for the study of animal behaviour and pathology. Please see the following articles: https://doi.org/10.3390/rs12213542;  

https://doi.org/10.1111/csp2.239). 

Line 182. in which way? Did you use a softweare? Please describe it. 

Conclusions

This part is quite evident, describing a situation that you can find in different areas of the world. For this reason is quite generic and I advice you giving pratic solutions adoptable in the study area. 

Reviewer 2 Report

Comments and Suggestions for Authors

It was interesting to read this paper about the impact of livestock predation by large carnivores in a province in Pakistan. The authors have tried to gain an understanding of the impact of predation on the people living in the region, and the way that these people view predators. I find it an interesting topic, and read the paper with great interest.

I do have some major concerns about this paper.

-        The English language needs a lot of work in this document. Some sections read really well, others not so much. In the detailed comments below, I have indicated many instances where the grammar, use of words, or phrasing of sentences needs to change, and suggested the changes needed, but stopped doing that after the study area section of the methods.  I would very strongly recommend that the authors find somebody proficient in the English language and with experience in writing scientific papers to proof-read this document.

-        The methods section needs a lot of work to clarify how the authors actually performed their research.

-        The discussion needs major improvements.

-        A study like this would need appropriate ethics permits. Other than a statement about a permit from the Forest and Wildlife Department, I do not see any information about this. This info needs to be added.

In addition, specific comments follow below:

Introduction

Page 2, lines 54-56. As human..interactions. The first part of that sentence is unclear. I doubt that the increasing number of livestock is the reason that the human population is increasing; it would be the other way around, however, that is the way this part of the sentence reads. Please re-phrase, so causation is turned around. Also, the relationship between people and animals. This is unclear as well; livestock are also animals. I assume the authors mean wild animals here? If so, please change.

Page 2, line 60. Delete compelling

Page 2, line 62. Replace nevertheless with however

Page 2, line 67. Replace deadly predator control methods with lethal predator control methods.

Page 2, line 68-69. Delete the sentence these actions. was intended.

Page 2, line 69-71. they.biodiversity. Replace they threaten with these threaten. Replace species that were not intended with non-target species. Maybe the authors mean to use a variety of non-target species?

Page 2, line 71. Delete although. Also, ancient times is too vague a statement.

Page 2, line 78. Delete been observed with, the sentence part then reads: grey wolves have home ranges.

Page 2, lines 78-82. This section does not make sense. The authors give a range for wolf home-ranges and than continue quoting other studies with different home range sizes for wolves, that do not fall in range they just quoted. Please re-write this section.

Page 2, line 83. bears have been seen with a home range is not correct phrasing for this sentence. It is more correct to state bears have a home range…’.

Page 2, line 83-85. Similar comment for the bear ranges as for the wolves. The authors give an estimate for bear home range sizes, and then come up with different numbers. It does not make sense, please re-write.

Page 2, line 91-93. The male and female caracal ranges, are they winter or summer? The authors just stated there were large seasonal differences..

Page 2, 78-93; the section about home range sizes for different species. I think the authors are spending too much time going into detail about home-range sizes for different predatory species, which distracts from the flow of the introduction. It would be better to significantly shorten this section. The authors could structure it for example by stating which predators occur in the region, and then giving an estimate of the largest home range size (and the species it would belong to), and the smallest one (and the species it would belong to). This is interesting and valuable information, but as it is written at the moment, this section has too much detail and distracts from the message the introduction is trying to give.

Methods

Page 3, study area. It would be good to have a map showing the study area, putting it in context of its geographical location.

Page 3, line 120. Delete in terms of its climate

Page 3, line 122. Delete in contrast

Page 3, line 123. Replace while January is with with January being.

Page 3, line 139-139. questionnaires study area This sentence does not give any useful information. It raises more questions than it answers, which is never good in the methods. Please delete, I am sure the individual methods would be explained further down in this section.

Page 3, line 139-140. Prior Department. Ok, but for what?

Page 3, line 143. that were rife with. Emotive language, please change.

Page 3, line 145-146. such as within the study area. This is not strictly necessary to mention again, given the list of predators in the region is already mentioned under the study area.

Page 4, line 150. each potential research site within these grids, ok, but what constitutes a potential research site?

Page 4, line 151. I would strongly recommend not using hamlet, call them maybe villages?

Page 4, line 151. How were the 10 participants in each village selected?

Page 4, line 154. It is not correct to end a sentence with ‘…., and.

Page 4, line 156. as documented by (reference) is unclear. Did the reference already collect all this data?

Page 4, lines 156-158. to ensure ..confidential. This is unclear. Do the authors imply that the information given by younger people is not as accurate or comprehensive as from older people? And that younger peoples personal information was not kept confidential?

It feels to me that the 2.2 methods sub-section is not finished. The authors leave the section of after saying they did a preliminary survey, then the section ends. What about the proper, real survey? Or do the authors mean to say that they started their research efforts with the survey, before using other methods? Either way, clarification is needed here.

Page 4, line 165. Oh I found the study area map. That is great, but the text needs a reference to this figure. Also, from this map, it is not clear where in Balochistan the study area is.

Page 4, line 169-160. normally.survey results. This sentence is unclear. It is not proper English, and having read it over and over again I cannot guess what the authors are trying to say with this.

Page 4, section sign surveys and occupancy analysis
This section needs a lot of work. The use of English language is very poor, and I had to re-read the whole section multiple times to get some idea of what the authors were trying to describe. Please re-write this section. Also, please explain how multiple villagers can create a repeat history for sightings of a species, as generally repeat sightings are temporally segregated.
It is not enough to state that site and survey covariates were extracted. Which covariates? How were they extracted? That needs a lot more detail. Also, what signs were used?
Also, there are no results presented about an
occupancy analysis from McKenzie, which is what I expected. Where are those results, or if that was not the analysis undertaken, please re-write this section to reflect that.

Page 4, line 183. Data analysis. What software was used in this analysis?

Page 4, line 184-185. Weregion. Ok, but occurrence and intensity of livestock depredation based on what data?

Formula 1. Ok for the formula, but if that is the model that was used, then that does not match the text above, stating what the authors were looking at. Where in this formula is occupation, predator type or economic losses due to depredation?

Formula 2. Same comment as above, it does not reflect the text above. I get the feeling that the authors got the formulas mixed up (1 should be 2, 2 should be 1), but even then not all the factors mentioned in the text are reflected in the models.

Regarding the models that the authors use, how were they used? Were they simply used to test for significance of effects within the model? This is not very reliable with general linear regression, and not a good statistical data analysis. It would be better to come up with a hypothesis of which variables potentially influence the dependent variable, and then run different models, testing the different hypothesis. Model ranking (for example AIC) can then be used to pick the model that best describes the data, and the effect of the explanatory variables can then be presented in the results.

Results

Table 1. The authors need to check their percentages. In different categories that needs to add up to 100, it does not always do that, which is impossible, or needs explanation.

Page 5, line 204-206. a small educated. Simply because they are aware of carnivore conservation, does not necessarily mean they are educated. If the authors made the that connection from their own data, that needs to be clearly indicated.

Figure 2 is not very clear. It might be more useful (and more intuitive to understand) if the authors use a stacked column chart for this, where, within the one graph, each livestock type has one column, and each predator type gets its percentage allocated within that column.

Page 8, line 235-237. reference to . carnivores. This is unclear, what do the authors mean by this?.

Page 8, line 240-241. future perceptions. This sentence is unclear, I am not sure what the authors are trying to say here.

Page 8, line 244. this. these animals. Ok, but that is not a result, that should be discussed in the discussion.

Figure 3. Needs a better explanation in the caption about what the percentages are. Graphs and Tables need captions that allow the reader to understand the figure by itself, without having to read the text of the paper to understand it.

Figure 4. Needs a better caption; what are the numbers above the bars?

Figure 5. Needs a more descriptive caption.

Page 10, lines 281-286. a sign . Black bear. What is the difference between these results and the results on page 9, under 3.6 Sign occupancy survey …’ ? This is confusing.

Figure 6. Needs a more descriptive caption.

Discussion

-        Figure 7. I am not sure what figure 7 adds to the discussion. It is not necessary there.

-        A discussion is not a place to present new results (Table 5 and associated explanation in the text belongs in the results section) and not a place to refer back to figures used in the results section (figure 6 is referred to).

-        The discussion needs a lot more structure. The authors needs to pick the main points from their results they want to discuss, and have a clear and concise discussion about these points, ground them in the existing literature and compare and contrast their results with findings from other studies.  At the moment different topics are referred to in different parts of the discussion, often coming back to the same topic later on. It reads very jumbled, confusing and unclear.

-        It is often unclear whether the authors are talking about their own results or the results from another study they are referring to. This needs more clarification throughout the whole discussion.

There is a lot of reference in the whole paper about retaliatory killing of predators. However, the authors do not collect data on this. Where does this information come from that this happens?

Comments on the Quality of English Language

-        The English language needs a lot of work in this document. Some sections read really well, others not so much. In the detailed comments I have indicated many instances where the grammar, use of words, or phrasing of sentences needs to change, and suggested the changes needed, but stopped doing that after the study area section of the methods.  I would very strongly recommend that the authors find somebody proficient in the English language and with experience in writing scientific papers to proof-read this document.